# Augmentation of EPR Effect and Efficacy of Anticancer Nanomedicine by Carbon Monoxide Generating Agents

**DOI:** 10.3390/pharmaceutics11070343

**Published:** 2019-07-16

**Authors:** Jun Fang, Rayhanul Islam, Waliul Islam, Hongzhuan Yin, Vladimir Subr, Tomas Etrych, Karel Ulbrich, Hiroshi Maeda

**Affiliations:** 1Faculty of Pharmaceutical Sciences, Sojo University, Kumamoto 860-0082, Japan; 2Department of Microbiology, Graduate School of Medical Sciences, Kumamoto University, Kumamoto 860-8556, Japan; 3Biodynamics Research Foundation, Kumamoto 862-0954, Japan; 4Department of General Surgery, Sheng Jing Hospital, China Medical University, Shenyang 110011, China; 5Institute of Macromolecular Chemistry, Czech Academy of Sciences, Prague 16206, Czech Republic

**Keywords:** EPR effect, nanomedicine, carbon monoxide, PDT, nanoprobe

## Abstract

One obstacle to the successful delivery of nanodrugs into solid tumors is the heterogeneity of an enhanced permeability and retention (EPR) effect as a result of occluded or embolized tumor blood vessels. Therefore, the augmentation of the EPR effect is critical for satisfactory anticancer nanomedicine. In this study, we focused on one vascular mediator involved in the EPR effect, carbon monoxide (CO), and utilized two CO generating agents, one is an extrinsic CO donor (SMA/CORM2 micelle) and another is an inducer of endogenous CO generation via heme oxygenase-1 (HO-1) induction that is carried out using pegylated hemin. Both agents generated CO selectively in solid tumors, which resulted in an enhanced EPR effect and a two- to three-folds increased tumor accumulation of nanodrugs. An increase in drug accumulation in the normal tissue did not occur with the treatment of CO generators. In vivo imaging also clearly indicated a more intensified fluorescence of macromolecular nanoprobe in solid tumors when combined with these CO generators. Consequently, the combination of CO generators with anticancer nanodrugs resulted in an increased anticancer effect in the different transplanted solid tumor models. These findings strongly warrant the potential application of these CO generators as EPR enhancers in order to enhance tumor detection and therapy using nanodrugs.

## 1. Introduction

Enhanced permeability and retention (EPR) effect is the basic concept for the tumor-targeted delivery of macromolecular anticancer agents [1,2]. It is based on the unique physiological and anatomical nature of tumor tissues, which show as high vascular permeability and defected lymphatic clearance, and consequently, the progressive accumulation of macromolecules into tumor tissues is observed [1,2]. Many anticancer nanodrugs have been developed based on the EPR effect that show a superior tumor targeting effect and therapeutic activities [3,4,5,6,7]. However, the concept of the EPR effect was primarily based on small size/early stage solid tumors in mice. In contrast, many cancers seen in clinics are at an advanced stage and are of a large size. The blood vessels of such tumors are often occluded or embolized [8,9,10,11]. Also, the advanced large tumors have many necrotic areas or degenerated blood vessels, and tumor blood flow is always irregular, so one can claim that the EPR effect also exhibits a high degree of heterogeneity [5,6]. Because of such suppressed blood flow as seen by angiography, drug delivery to tumors is frequently very poor. As a matter of fact, many refractory tumors, such as pancreatic cancer and metastatic liver cancer, have poor blood flow, thus even nanodrugs or sophisticated antibody drugs cannot be effectively delivered into the tumor. Thus, the challenges to open-up tumor blood vessels and restore tumor blood flow are critical and of great importance for anticancer nanomedicine. 

Based on this knowledge, we have developed a method counteracting against the problems by using a vascular dilator, nitric oxide (NO) donor such as nitroglycerin (NG), and an angiotensin-I converting enzyme (ACE) inhibitor such as enarapril [5,6,12,13]. We and other groups have further described additional strategies to enhance the EPR, and thus the therapeutic effect of anticancer nanodrugs using nano-designed NO donors [14,15]. More recently, we further elaborated various NO generating agents, all of which are clinically used drugs, and obtained a significantly increased antitumor effect when combined with the proto type nanodrugs (e.g., polymer conjugated pirarubicin—P-THP), in five different tumor models, including carcinogen-induced autochthonous advanced tumors of the colon and breast [8]. The results definitely showed the usefulness of these EPR enhancers.

Along this line, in this study, we focused on a new potential EPR enhancer, carbon monoxide (CO). CO is mostly generated in the body during heme degradation, catalyzed by heme oxygenase (HO), and is well known to play versatile roles including similar functions to that of the vasodilation of NO [16,17,18]. Previously, we investigated the effect of an HO inducer pegylated hemin (PEG-hemin), and tricarbonyldichlororuthenium(II) dimer (CORM2), which is a CO-releasing molecule, on the vascular permeability and tumor accumulation of polymer drugs [19]. Meanwhile, we successfully developed a polymer micelle of CORM2 using a styrene maleic acid copolymer (SMA/CORM2), which showed a prolonged plasma half-life and sustained release of CO, thus exhibiting a targeting property to inflammatory and tumor tissues by virtue of the EPR effect [20]. These nano type CO generators are suitable for in vivo application, thus showing their potential as new EPR enhancers.

Here, we examined the applicability of these polymeric CO generators as EPR enhancers for the recently developed polymeric nano photosensitizer (PS) designed for photodynamic therapy (PDT), (i.e., pHPMA [*N*-(2-hydroxypropyl)methacrylamide copolymer]-conjugated pyropheophorbide-a (P-PyF)) [21], which demonstrated a high tumor accumulation and favorable tumor imaging. Furthermore, pHPMA-conjugated pirarubicin (P-THP), which shows excellent antitumor effects against various solid models [22], was also used to confirm the effect of CO generators.

## 2. Materials and Methods 

### 2.1. Materials

We synthesized P-PyF [21] and P-THP [22], as previously described. The *Mw* of P-PyF was about 31,000 g/mol, and in a water solution, it forms micelles with a mean particle size of 194 nm; P-THP has an apparent *Mw* of 40,000, and a hydrodynamic diameter of 8.2 nm. RPMI 1640 medium, Dulbecco’s modified eagle Medium (DMEM), isoflurane, Evans blue, and all of the solvents were purchased from Wako Pure Chemical (Osaka, Japan). CORM2, hemin, and poly(styrene maleic anhydride) [SMA] were from Sigma-Aldrich Chemical (St. Louis, MO, USA). Succinimidyl derivative of poly(ethylene glycol) [PEG], with an average molecular weight of 2000, was purchased by NOF Co., Tokyo, Japan. PEG-hemin [23] and SMA/CORM2 [20] were prepared as reported previously. Fetal bovine serum was obtained from Nichirei Biosciences Inc. (Tokyo, Japan). The in vivo fluorescence imaging probes, oxygen-doped single-walled carbon nanotubes (o-SWCNT-PEG) [24], were kindly provided by Shimadzu corporation (Kyoto, Japan). The other reagents and solvents of a reagent grade were from commercial sources and were used without further purification.

### 2.2. Animals, Cells, and Tumor Models

Male ddY mice, male Balb/c mice, and male C57/BL mice (all six weeks old) were purchased from SLC, Shizuoka, Japan. All of the animals were maintained at 22 ± 1 °C and a 55% ± 5% relative humidity, with a 12-hour light/dark cycle. All of the experiments were approved by the Animal Ethics Committees of Sojo University (no. 2018-P-022, approved on April 01, 2018), and were carried out according to the Laboratory Protocol for Animal Handling of Sojo University. 

The mouse sarcoma S180 cells, maintained by a weekly passage in ascitic form in the ddY mice, were implanted subcutaneously (2 × 10^6^) in the dorsal skin of the ddY mice. Mouse C26 colon cancer cells and melanoma B16-F10 cells were maintained in in vitro cultures by using RPMI-1640 and DMEM, respectively, both supplemented with 10% fetal bovine serum under 5% CO_2_/95% air at 37 °C. The cultured cells were collected and suspended in physiological saline to a concentration of 2 × 10^7^ cells/mL, and 0.1 mL of cell suspension was implanted into the dorsal skin of Balb/c mice and C57/BL mice, to obtain solid tumors, respectively. In different experiments, various numbers of mice (three to eight) were used in different experiment group. 

### 2.3. Generation of CO in Tumors and Normal Tissues after Administration of CO Generators

The C26 tumor implanted mice were used to investigate CO generation after the intravenous (i.v.) administration of PEG-hemin or SMA/CORM2. The experiments were performed 12–14 days after tumor inoculation, when the tumors grew to a relatively large size of 10–15 mm in diameter. PEG-hemin (10 mg/kg, hemin equivalent) and SMA/CORM2 (10 mg/kg) were injected i.v., by these doses significant increases of CO in circulation, and tumor and inflammatory tissues were achieved [19,20]. After a scheduled time (i.e., 2, 4, 8, and 24 h), the mice were killed and the tumors and normal tissues (e.g., blood, kidney, and liver) were collected. To 100 mg of each tissue, 0.4 mL of 0.01 M phosphate-buffered 0.15 M saline (PBS) was added and homogenized by A Polytron^®^ homogenizer in the test tubes. To 0.5 mL of homogenized tissue in 10-mL glass test tubes, 0.2 mL of saponin (16.7 mg/mL) was added to disrupt SMA/CORM2 micelles, and to release CO into the tissue [25]. The tubes were then sealed tightly with silicon rubber stoppers. After 24 h, 1 mL of the gas in the test tubes was collected by syringe and subjected to gas chromatography (TRIlyzer mBA-3000; Taiyo Instruments, Inc., Osaka, Japan), as described previously [20].

### 2.4. Augmentation of the Tumor Accumulation of Nanodrugs by Using CO Generators

In the C26 tumor model, for control of the no EPR enhancer group, P-PyF of 5 mg/kg (PyF equivalent) dissolved in PBS was injected i.v. via the tail veins. At 24 h after injections, the mice were killed and blood was withdrawn from the inferior vena cava, followed by perfusion with 20 mL of PBS. The tumors and normal tissues were then dissected and weighed, and DMSO (1 mL/100 mg of tissue) was added. The tissues were then homogenized. After centrifugation of the samples (12,000 rpm, 15 min), P-PyF in the supernatants were quantified by fluorescence intensity at ex.390 nm/em 680 nm. 

For the EPR enhancer treated groups, PEG-hemin was injected i.v. 24 h before, and SMA/CORM2 was injected i.v. 2 h before the i.v. injection of P-PyF. In some experiments, the mice were subjected to in vivo imaging using an IVIS XR system (Caliper Life Science, Hopkinton, MA, USA) after the above-described protocol, to visualize the tumor accumulation by fluorescence of this polymeric nanoprobe.

### 2.5. In-Vivo Therapeutic Effect of Nanodruges in Combined with CO Generators

In the S180, C26, and B16 tumors, 10–12 days after tumor injection when the tumors reached diameters of about 10 mm, the P-PyF dissolved in PBS at indicated concentrations (high dose and low dose) was administered i.v., and CO generators were administered in combination with a low dose of nano-probe, which were injected 24 h (PEG-hemin) or 2 h (SMA/CORM2) before the administration of P-PyF, as described above. At 24 and 48 h after the administration of P-PyF, the tumors were irradiated via an endoscopic fiber optics system (MAX-303; Asahi Spectra, Tokyo, Japan) with xenon light at 400–700 nm for 5 min (36 J/cm^2^). In separate experiments, a combination therapy of P-THP with SMA/CORM2 as a reference control of the chemotherapy was also carried out by similar protocol as described above, in which P-THP (5 mg/kg, THP equivalent) was injected i.v. only once 2 h after the administration of SMA/CORM2. Tumor volume (mm^3^) was calculated as (W^2^ × L)/2 by measuring the tumor width (W) and length (L), and the body weights of the mice were measured during the study period. 

We also investigated the suppression of lung metastasis in the C26 tumor inoculated mice, which will become apparent 30–40 days after tumor inoculation at the dorsal skin [26,27]. Then, after about 40 days after tumor injection, the mice were killed and the lungs were excised. The number of metastatic nodules on the surface of the lung were counted visually, and the cumulative size of the metastatic nodules was calculated by the addition of the diameters of each tumor nodule.

### 2.6. In Vivo Imaging of Tumor Blood Vessels

In this study, the in vivo imaging of tumor blood vessels (angiography) of the S180 tumor was performed with the use of a fluorescence imaging probe, o-SWCNT-PEG, in a portable in vivo fluorescence imaging system (SAI-1000, Shimadzu Corporation), with mouse under anesthesia, using isoflurane.

### 2.7. Statistical Analyses

All of the data are expressed as means ± standard deviation (SD). The data were analyzed using analysis of variance (ANOVA), followed by the Bonferroni correction. A difference was considered statistically significant when *p* < 0.05.

## 3. Results

### 3.1. Tumor-Selective Generation of CO by SMA/CORM2 and PEG-hemin 

As Figure 1 shows, the liver showed the highest CO concentration, followed by the tumor. However, it should be noted a very high level of CO at 0 h (without SMA/CORM2 treatment) in the liver is not coming from SMA/CORM2, but contributes to the endogenous HO that is dominantly present in the liver, as it is the organ of heme degradation. After an i.v. injection of SMA/CORM2, the CO concentration in the tumor increased remarkably, which reached about 10 times that of the untreated control tumors (0 h) after 2 h, and it declined gradually but remained relatively high up until 24 h. We also found significantly increased CO concentrations in the blood up until 2 h after the i.v. injection of SMA/CORM2 compared with the untreated control, and about a 1.5-time higher CO concentration was found in the liver after 4–8 h, but it was statistically insignificant; no apparent increases of CO amount in the other normal tissues were seen (e.g., kidney; Figure 1A). These findings indicated that CO generation is more selective in tumors by SMA/CORM2, because of the tumor preferred accumulation of this micelle by the EPR effect. These results are consistent with our previous study of SMA/CORM2 [20]. Accordingly, we administered SMA/CORM2 2 h before the nanodrugs in the following experiments.

In the case of PEG-hemin, which also accumulated in the tumor by an EPR effect, hemin is known to induce HO-1, that yielded CO in the tumor progressively in time (Figure 1B). At 24 h after injection of the PEG-hemin, a more than nine-time increase in the CO concentration in the tumor was achieved (Figure 1B), while a 2.5-fold increase in CO was found in the liver; 1.5 and 3-fold increases in the CO levels were observed in the blood and kidney respectively, however, the concentrations of CO were much lower than that in tumor (Figure 1B). Thus, an increase in the CO level by PEG-hemin is also accompanied with a relatively high tumor-selectivity. Based on this result, PEG-hemin was administered 24 h before the injection of nanodrugs in the following experiments.

### 3.2. Increased Accumulation of Polymeric Nanodrugs in Tumors by CO Generators

As shown in Figure 2A, the tumor accumulation of P-PyF in the C26 tumor model was higher than in most of the normal tissues, except for the liver and the spleen, which are rich in reticuloendothelial systems for capturing macromolecules [28], also take up porphyrin derivatives actively as the organs of heme degradation [21]. More importantly, after treatment with SMA/CORM2 or PEG-hemin, the accumulation of P-PyF in the tumor 24 h after i.v. injection increased 1.5~2 fold (Figure 2A). In vivo imaging also showed that a more intensified fluorescence image of P-PyF treated tumor was seen when compared with the control mice without pretreatment with CO generators (Figure 2B).

### 3.3. Improved Therapeutic Effects of Nanodrugs by CO Generators

We first examined PDT using P-PyF in the C26 tumor. The drug was injected only once, followed by xenon light irradiation at 24 and 48 h after P-PyF injection i.v.. As shown in Figure 3A, 2 mg/kg P-PyF (PDT-L) did not show a clear antitumor effect, whereas it showed a more marked suppression of the tumor at 5 mg/kg (PDT-H) by PDT. The combination therapy of PDT using 2 mg/kg P-PyF (PDT-L) with CO generators significantly augmented the PDT effect, resulting in almost the same therapeutic effect as that using 5 mg/kg P-PyF alone (PDT-H; Figure 3A). Moreover, this combination therapy also significantly reduced the lung metastasis in this model. Similar to the results shown in Figure 3A, PDT-L (low dose) did not decrease the lung metastasis apparently, but PDT-H (high dose) clearly reduced the numbers of metastatic nodules in the lung (Figure 3B). The pretreatment of PEG-hemin with PDT-L remarkably suppressed the number of metastatic nodules in the lung compared with PDT-L alone, which was comparable to PDT-H. Similar results were also found for pre-treatment with SMA/CORM2, although no statistic difference was obtained (*p* = 0.077; Figure 3B). These findings suggested that the CO generators did enhance the therapeutic effect of polymer nanodrugs by about two times, even for the metastasis. These findings clearly indicated that by combining with CO generators (EPR enhancer), a low dose of nanodrugs could achieve a similar effect as that of the two- to three-fold higher dose, which would not only increase the therapeutic efficacy, but also contribute to reducing undesirable side effects and treatment cost.

Similar findings were also observed in the murine melanoma B16 model, which is an aggressive tumor. We confirmed an improved therapeutic effect for the combination therapy of CO generators with nano PS, P-PyF. The results were similar to those shown in Figure 3A, in which a significant potentiation of the antitumor effect with CO generators was confirmed (Figure 4). 

We further proved this concept by using another polymeric anticancer drug P-THP, in combination with SMA/CORM2 in different solid tumor models in mice. As shown in Figure 5A, P-THP alone at 5 mg/kg suppressed the tumor growth of the C26 tumor, whereas a significantly augmented therapeutic effect was achieved when pretreated with SMA/CORM2. The lung metastasis in this model was also remarkably inhibited by this combination therapy (inset of Figure 5A). Also, we observed similar results in the S180 tumor that, in combination with SMA/CORM2, significantly improved the therapeutic effect of P-THP (Figure 5B).

In addition, in a separate experiment, the CO donors alone at above-described doses did not show significant antitumor effects (data not shown).

### 3.4. Restoration of Tumor Blood Flow by CO Generators

To analyze the possible mechanisms of the augmentation of the EPR effect as a result of CO generation, we employed angiography to reveal the vascular blood flow of the S180 tumor, using a fluorescence imaging system (SAI-1000). As Figure 6 illustrates, the use of both CO generators clearly enhanced the tumor blood flow, which could only weakly be seen without pretreatment with CO generators. These findings strongly indicated the improvement of tumor blood flow by the CO generators, consequently resulting in an augmentation of the EPR effect and therapeutic outcome.

## 4. Discussion

A lack of tumor selectivity is the major cause for the failure of anticancer chemotherapy. Most conventional anticancer drugs distribute indiscriminately in the body, not only to cancer tissues, but also more to the normal tissues, resulting in dose-limiting adverse effects, and consequently greatly decreasing the quality of life (QOL) of patients. The adverse effects in turn limit the dose escalation of anticancer agents, resulting in therapeutic failure. Therefore, a desperate need for tumor selective drug targeting has become a consensus in the field. 

In this regard, EPR effect-based nanomedicine is becoming a promising anticancer modality, which utilizes the anatomical and physiopathological natures of the tumor blood vessels. Unlike molecular targeting drugs aimed at the molecular or subcellular level, this approach primarily targets the tumor mass as a whole [2,5,6,29,30,31]. Because the rapid growth of tumors requires more nutrients and blood supply than normal tissues; incidentally, it shows unique vascular characteristics—extensive angiogenesis and immature, architecturally, and physiologically abnormal blood vessels. For these reasons, high vascular permeability and defected lymphatic recovery (EPR effect) are eminent. These factors end up in the accumulation and retention of macromolecules in tumor tissues, as observed and supported in many in vivo studies, as well as clinical studies [1,7,29,30,31]. These phenomena solely occur in solid tumors and inflammatory tissues, but not in normal tissues [29]. Thus, the EPR effect is the unique basic principle for the tumor targeting strategy using macromolecular drugs or polymer-based nanodrugs. 

Besides the anatomical abnormalities of the tumor blood vessels, we found that various vascular mediators, such as bradykinin, NO, vascular endothelial growth factor, and prostaglandins are extensively produced in tumor tissues, which are the major factors causing the EPR effect involving vascular permeability [32,33]. We indeed utilized these vascular mediators (e.g., NO generators and ACE inhibitors) to augment the EPR effect and thus enhance the delivery of nanodrugs to tumor tissues. 

Conventional low-molecular-weight drugs commonly accumulate in tumor tissues at less than 0.1, or a 1% range, of the injected drugs [1,34,35], whereas nanodrugs accumulate in tumor tissues at relatively high concentrations (i.e., 5%–10%) of the total injected drug, by taking advantage of the EPR effect [1,34,36,37]. Namely, 10–50 times higher concentrations of drugs can be delivered into tumor tissues 24–72 h after the i.v. injection of nanodrugs [1,34]. Ample studies, including our studies, have been exploiting tumor targeted nanomedicine based on EPR effect [3,4,5,6,7]. The augmentation of the EPR effect will therefore further enhance the tumor delivery and therapeutic effect of anticancer nanodrugs, promoting the development of anticancer nanomedicine. 

Furthermore, Navi et al. and other groups have demonstrated that the potential of thrombogenic and clot forming level in the tumor vasculature is highly elevated. As a result, vascular blood flow is so frequently obstructed just like cardiac infarct [9,10,11]. That means no blood flow and no drug delivery. For this reason, vascular dilation and the restoration of blood flow is critical for better cancer chemotherapy using an enhancer of the EPR effect in nanomedicine, which is also applicable for polymeric PS for PDT or radiotherapy.

Accordingly, in our laboratory, we first showed that NO could increase vascular permeability and blood flow [12]. More recently, we successfully established a combination therapy using NO donors such as nitroglycerin, L-arginine, and hydroxyurea [8], as well as an ACE inhibitor that potentiates bradykinin [14], to augment the anticancer efficacy of nanodrugs. In the present study, we specifically focused on CO, which is a gaseous molecule having similar functions as NO, in order to enhance blood flow and vascular permeability [16,17,18]. Here, we selected two CO-generating agents, one is CORM2 micelle (CO donor) that generates CO by decomposition of the micelle when given extrinsically [20]; another agent is hemin, which is the inducer of HO-1 in vivo that will generate CO endogenously during the hemin degradation pathway [23]. Our previous study using CORM2 and polymer conjugated hemin (PEG-hemin) showed the increased tumor delivery of macromolecules (Evans blue bound albumin) by these CO generators, by virtue of the improved EPR effect and the restoration of the tumor blood flow [19]. However, the therapeutic effect was not examined. Therefore, here, we further present the potential of the CO generators as EPR enhancers to reinforce the antitumor effect of nanodrugs, including PDT, using polymeric PS.

One critical and important issue for EPR enhancers is that they must generate or release the mediators (e.g., NO and CO) selectively in the tumor. The above-described NO donors all show a tumor-specific production of NO by different mechanisms [8]. The CO generators used in this study (SMA/CORM2 and PEG-hemin) are polymeric micelle, which showed a prolonged circulation time and preferred accumulation in tumor tissues [19,20,23]. Consequently, a tumor-selective production of CO could be achieved (Figure 1), which thus induced the increase in the vascular permeability and restored tumor blood flow, as revealed by the angiography (Figure 6), which is consistent with the results of our previous study [19]. The CO production in tissues induced by these CO generators is within the physiological levels (~10 ppm), which are far lower than the toxic dose of CO (e.g., >200 ppm), which ensures the safety of use of these CO generators. 

The use of CO generators resulted in an augmented tumor delivery of nanodrugs, which was about 1.5–2 times greater than that without CO generators (Figure 2), while no apparent effect on the normal tissue was seen (Figure 2). This effect was attributed to the tumor-specific production of CO (Figure 1). 

Consequently, the therapeutic effect of nanodrugs, that is, the polymer-conjugated anticancer drug P-THP and the polymeric PS, P-PyF used for PDT, were confirmed. Namely, a combination treatment of these nanodrugs with CO donors in three implanted murine tumor models (i.e., S180, C26, and B16) resulted in a two- to three-fold improved therapeutic effect for PDT using P-PyF (Figure 3 and Figure 4); combination chemotherapy with P-THP also resulted in about a two- to three-times better therapeutic effect than the P-THP monotherapy (Figure 5). This improved therapeutic effect was not only seen in the implanted tumors, but also observed in suppression of metastasis (Figure 3B). These improved results will be attributed to increased drug accumulations in the tumor by the EPR effect (Figure 1). Accordingly, studies of developing anticancer nanomedicine via the EPR effect, which is a promising approach for tumor targeting [3,4,5,6,7], will be largely encouraged by using these EPR enhancers, such as CO generators. 

In the present study, we also confirmed the restoration of the tumor blood flow by the angiographic technique using a nano-fluorescence probe, which could clearly visualize the blood vessels in the tumor (Figure 6). In this study, we used relatively large size tumors as the model for advanced stage tumors. Under these circumstances, more tumor blood flow became visible in the tumor region (Figure 6). 

In addition to the improvement of the tumor blood flow, the Wegiel and Otterbein’ group also reported that CO could induce apoptosis in lung cancers by altering the tumor microenvironment to increase the CD86 expression, and then activating mitogen-activated protein kinase (MAPK)/extracellular signal-regulated kinases (Erk) 1/2 pathway [38]. The inhibition of tumor growth by CO was also found in prostate cancer cells by targeting the mitochondria activity to expedite the metabolic exhaustion of cancer cells [39]. Such multiple mechanisms thus may act on the improved antitumor effect of nanodrugs by CO donors, which warrants additional investigations.

## 5. Conclusions

We investigated the potential of the tumor selective CO generators as an enhancer of the EPR effect, so as to improve the therapeutic effect of nanodrugs. By using different solid tumor models, we demonstrated increased tumor accumulation of nanodrugs when combined with CO generators (i.e., SMA/CORM2 and PEG-hemin; Figure 2). The increased tumor accumulation of nanodrugs consequently resulted in an improved therapeutic effect (Figure 3, Figure 4 and Figure 5). These findings strongly suggested the importance of an enhancement of EPR effect, and the applicability of CO generators as similar to NO donors. We emphasize the importance of the tumor-selective enhancement of the EPR effect for clinical application.

## Figures and Tables

**Figure 1 pharmaceutics-11-00343-f001:**
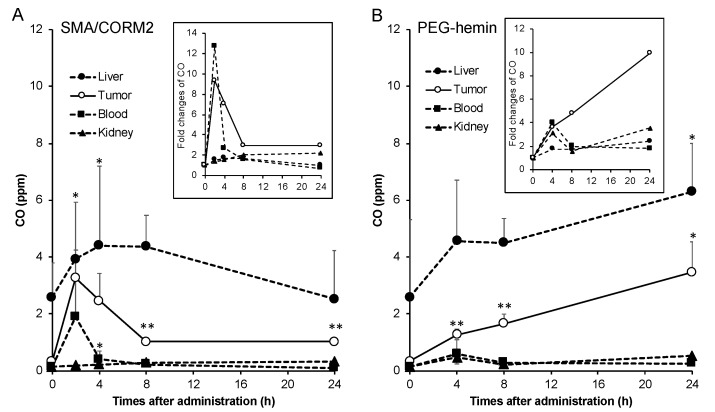
Generation of CO in the tumor and normal tissues after administering CO generators. Mice bearing colon C26 tumors (diameters of 10–15 mm) were used. SMA/CORM2 (**A**) and PEG-hemin (**B**) were injected i.v., and after a scheduled time, the mice were killed and the tumors and normal tissues were collected. To 100 mg of each tissue, 0.4 mL of a 0.01 M phosphate-buffered 0.15 M saline (PBS) was added and homogenized in test tubes, and 200 μL of saponin was then added. After 24 h of incubation at room temperature, CO in the test tube was quantified by gas chromatography. The inset shows the fold changes of CO in the tissues. *, *p* < 0.05; **, *p* < 0.01. The data are means ± standard deviation (SD), *n* = 3–4. See text for details.

**Figure 2 pharmaceutics-11-00343-f002:**
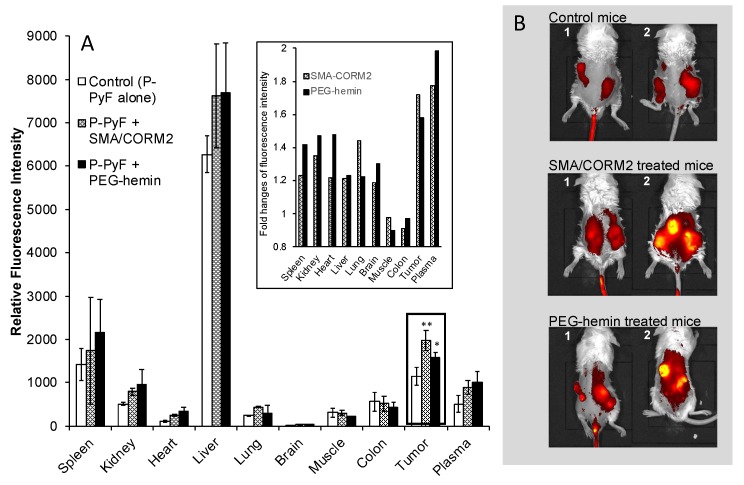
Effect of CO generators on the body distribution of pHPMA [*N*-(2-hydroxypropyl)methacrylamide copolymer]-conjugated pyropheophorbide-a (P-PyF) in C26 tumor-bearing mice. PEG-hemin was injected i.v. 24 h before, and SMA/CORM2 was injected i.v. 2 h before the administration of P-PyF (5 mg/kg, PyF equivalent). At 24 h after P-PyF administration, two mice in each group (1 and 2) underwent imaging by an IVIS in vivo imaging system (**B**), followed by the quantification of a P-PyF amount in each tissue by detecting the fluorescence from PyF (**A**); the inset of (**A**) shows the fold changes of fluorescence from PyF. *, *p* < 0.05; **, *p* < 0.01. The data are means ± SD, *n* = 3–6. See text for details.

**Figure 3 pharmaceutics-11-00343-f003:**
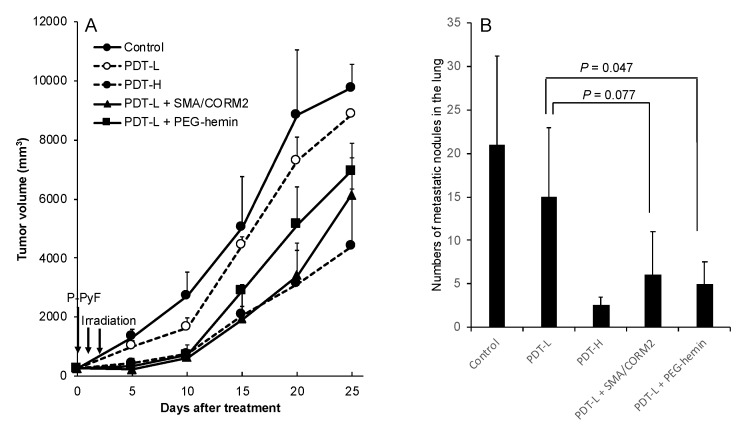
Enhanced therapeutic effect of P-PyF and CO generators under the photodynamic therapy (PDT) method on C26 tumors. Two concentrations (2 and 5 mg/kg, PDT-L and PDT-H, respectively) of P-PyF were injected i.v. when the tumor diameters were measured as about 8–10 mm. PEG-hemin and SMA/CORM2 were injected i.v., respectively. Then, 24 and 2 h later, P-PyF was injected i.v., followed by light irradiation (120 mW/cm^2^, 5 min, 36 J/cm^2^). Tumor growth was measured every five days. (**A**) Original implanted primary tumor and (**B**) metastatic tumor nodules in the lung were suppressed by PDT dose-dependently, and low-dose PDT, when combined with CO generators, exhibited a similar antitumor effect as high-dose PDT. Data are means ± SD; *n* = 4–8. See text for details.

**Figure 4 pharmaceutics-11-00343-f004:**
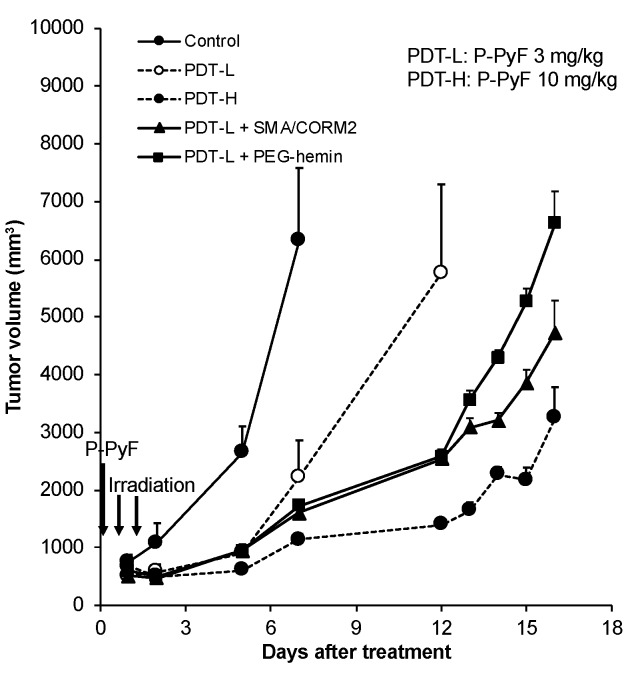
Enhancement of therapeutic effect of PDT using P-PyF in combination with CO generators on B16 tumors. When the tumor diameters reached about 10 mm, P-PyF (3 and 9 mg/kg) was injected i.v. PEG-hemin and SMA/CORM2 were administered in the same way as the protocol in Figure 3. After 24 and 48 h of P-PyF injection, the tumors were irradiated by a xenon light (120 mW/cm^2^, 5 min, 36 J/cm^2^). Data are means ± SD; *n* = 4–8. See text for details.

**Figure 5 pharmaceutics-11-00343-f005:**
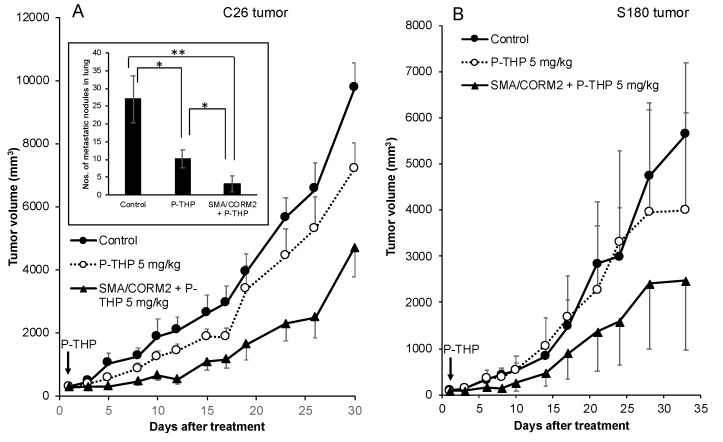
Augmentation of therapeutic effect of P-THP by SMA/CORM2 against C26 (**A**) and S180 (**B**) solid tumors. P-THP was injected i.v. only once at 5 mg/kg. SMA/CORM2 was applied 2 h before the P-THP injection. Inset of (**A**) showed the numbers of metastatic nodules in the lung. Data are means ± SD, *n* = 4–6. See text for details. *, *p* < 0.05; **, *p* < 0.01.

**Figure 6 pharmaceutics-11-00343-f006:**
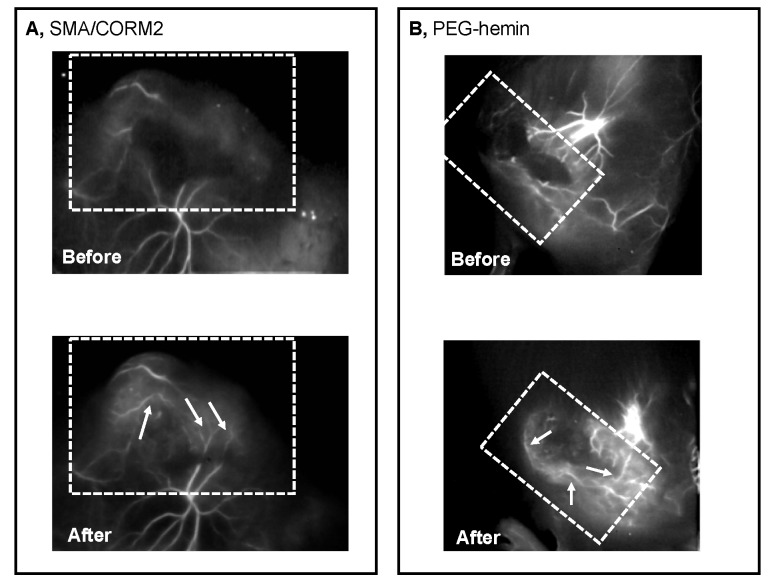
Improvement of tumor blood flow by SMA/CORM2 (**A**) and PEG-hemin (**B**), as indicated by fluorescence angiography. The S180 tumor-bearing mice with tumors of 15–20 mm in diameter were examined, and the angiography were performed immediately after the i.v. injection of the fluorescence nanoprobe using a SAI-1000 in vivo imaging system. After 24 h of nanoprobe injection i.v., SMA/CORM2 (10 mg/kg) or PEG-hemin (10 mg/kg, hemin equivalent) was administered i.v. After a scheduled time (2 and 24 h, respectively), the fluorescence nanoprobe was applied and an angiogram was obtained as described above. The dashed lines indicate the approximate area of the tumor, and the arrows indicate fluorescently visualized tumor blood vessels. The visuality of the blood vessels is very weak before the injection of CO generators (Before in (**A**) and (**B**)), compared with those after infusion of CO generators (After in (**A**) and (**B**)). See text for details.

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
