# Peer review of "Augmentation of EPR Effect and Efficacy of Anticancer Nanomedicine by Carbon Monoxide Generating Agents"

_pharmaceutics, 2019, doi:10.3390/pharmaceutics11070343_

Round 1

Reviewer 1 Report

In this paper, the authors have shown novel CO generating agents for the augmentation of EPR effect and its impact on the therapeutic efficacy of the nanomedicines. I have some minor issues that should be addressed:

1) Could the authors provide some physicochemical characterisation of nanomedicines?

2) Could the authors give reasons as to why nanomedicines such as 'Doxil'  was not used?

3) The authors should not describe large tumours as clinically relevant tumours 

Reviewer 2 Report

I find the paper very interesting. Its scope matches the topics covered by the journal. However, some essential changes are required. After they are applied, I recommend this paper for publication.

·         p.1, line 22, please delete “i.e” since you specifically used CO, the same in p.2, line 73

·         p. 2, line 95 – please provide the permission number for the animal study

·         p.2, 2. Materials and Methods, 2.2 Animals – please add the number of animals. Moreover, I can’t understand why the number of animals indicated in the figures legends varies i.e.  3-6, 3-4, 4-8. Please explain

·         p. 3, line 143 – “Then at about 40 after...” – I believe the word “days” is missing

·         p.3, lines 143-147. Please rephrase

·         p.5, line 208 – p.6, line 210, please rephrase. As shown in Figure 3B, PDT-H clearly reduces the number of metastatic nodules in the lung.

·         p.8, line 266 – please define QOL

·         p.9, line 219 – “By these reasons,….”, please correct. Things happen for a reason, not by a reason

·         p.9, line 293 – You say “..enhancers of the EPR effect are NOT only useful…” so the reader expects to read where else the enhancers are used. I believe the syntax is incorrect. Please check

Moreover,

·         You refer to “nanomedicines’ but I believe you mean “nanodrugs”

·         The paper would benefit most from the addition of figures representing the CO fold changes (in Fig. 1 and 2)

·         You need to discuss your results in relation to other studies that have exploited tumour targeting via the EPR effect. Please elaborate on this subject

Reviewer 3 Report

the authors presented a research investigating the potential of carbon monoxide (CO) as an EPR enhancer.  they used polymeric CO generators and showed they positive effect on accumulation of nanomedicine at tumor site due to enhanced EPR effect, and consequently increased therapeutic effect.

I think this paper is very well structured, has a solid rationale, and a good developed  methodology which leads to clear and well documented results. the group of the authors already have publications in the same field and this is a step forward in the direction of their research area.

Only few comments.

1) fig. 3A. treatment of tumors with PDT-L+SMA/CORM2 showed a comparable tumor reduction to treatment with PDT-H: please comment and  support the choice of using the lower concentration together with Co enhancers, compared to using the higher concentration. 

2) the concentration of PEG-hemin and SMA/CORM of 10 mg/kg, was it chosen based on which criteria? previously published results? if  yes, please give a ref. in par. 2.3
